# Protein Disulphide Isomerase and NADPH Oxidase 1 Cooperate to Control Platelet Function and Are Associated with Cardiometabolic Disease Risk Factors

**DOI:** 10.3390/antiox10030497

**Published:** 2021-03-23

**Authors:** Renato Simões Gaspar, Tanya Sage, Gemma Little, Neline Kriek, Giordano Pula, Jonathan M. Gibbins

**Affiliations:** 1Institute for Cardiovascular and Metabolic Research, School of Biological Sciences, University of Reading, Reading RG6 6AH, UK; t.sage@reading.ac.uk (T.S.); g.little@pgr.reading.ac.uk (G.L.); n.kriek@reading.ac.uk (N.K.); j.m.gibbins@reading.ac.uk (J.M.G.); 2Institute for Clinical Chemistry and Laboratory Medicine, University Medical Center Eppendorf Hamburg, D-20246 Hamburg, Germany; g.pula@uke.de

**Keywords:** platelets, protein disulphide isomerase, NADPH oxidase, metabolic syndrome, redox biology

## Abstract

Background: Protein disulphide isomerase (PDI) and NADPH oxidase 1 (Nox-1) regulate platelet function and reactive oxygen species (ROS) generation, suggesting potentially interdependent roles. Increased platelet reactivity and ROS production have been correlated with cardiometabolic disease risk factors. Objectives: To establish whether PDI and Nox-1 cooperate to control platelet function. Methods: Immunofluorescence microscopy was utilised to determine expression and localisation of PDI and Nox-1. Platelet aggregation, fibrinogen binding, P-selectin exposure, spreading and calcium mobilization were measured as markers of platelet function. A cross-sectional population study (*n* = 136) was conducted to assess the relationship between platelet PDI and Nox-1 levels and cardiometabolic risk factors. Results: PDI and Nox-1 co-localized upon activation induced by the collagen receptor GPVI. Co-inhibition of PDI and Nox-1 led to additive inhibition of GPVI-mediated platelet aggregation, activation and calcium flux. This was confirmed in murine Nox-1^−/−^ platelets treated with PDI inhibitor bepristat, without affecting bleeding. PDI and Nox-1 together contributed to GPVI signalling that involved the phosphorylation of p38 MAPK, p47phox, PKC and Akt. Platelet PDI and Nox-1 levels were upregulated in obesity, with platelet Nox-1 also elevated in hypertensive individuals. Conclusions: We show that PDI and Nox-1 cooperate to control platelet function and are associated with cardiometabolic risk factors.

## 1. Introduction

Cardiometabolic risk factors, such as central obesity, hyperglycaemia, insulin resistance, hypertension and dyslipidaemia are associated with increased production of reactive oxygen species (ROS) [1,2]. There is also a higher risk of developing thrombotic events partly due to platelet hyperactivation, which is pivotal to thrombus formation [3]. Upon vascular injury, these anucleated cells become activated through adhesion receptors and secondary mediator responses to form thrombi and reduce blood loss. Of note, both ROS and enzymes that regulate ROS generation are key to platelet activation in health and disease [4]. The study of enzymes that regulate ROS production in platelets may therefore be exploited to decrease thrombotic events associated with cardiometabolic risk factors.

Protein disulphide isomerase A1 (herein referred to as PDI) is a redox-responsive enzyme and an important regulator of platelet function both in vitro and in vivo [5,6]. PDI is the archetype of a family of thiol isomerases, also referred to as thioredoxins, capable of catalysing reduction, oxidation and isomerization of disulphide bonds [7]. PDI itself is released upon platelet activation [8] and was found to bind to integrin β3 during thrombus formation and regulate the affinity of integrin α_IIb_β_3_ thereby contributing to thrombus formation [6,9]. Moreover, PDI is also capable of interacting through disulphide exchange with other platelet adhesion receptors, such as integrin α_2_β_1_ [10] and glycoprotein Ibα [11].

In vascular smooth muscle cells (VSMC), PDI has been shown to regulate the production of reactive oxygen species (ROS) through aiding the assembly of nicotinamide adenine dinucleotide phosphate (NADPH) oxidase 1 (Nox-1) [12]. Similar to PDI, Nox-1 has recently been shown to regulate platelet function [13]. This enzyme complex is responsible for the production of ROS and is expressed on the outer membrane of vascular cells, such as endothelial cells [14], vascular smooth muscular cells (VSMC) [15] and platelets [16]. Only Nox-1, Nox-2 and Nox-4 isoforms have been found in platelets [17] although the presence of Nox-4 remains a matter of debate [18]. Using both selective inhibitors and in vivo deletion of the Nox-1 gene, Vara et al. [19] observed that Nox-1 is essential for superoxide production, platelet aggregation and thrombus formation downstream of the receptor for collagen GPVI. Meanwhile, Nox-2 was relevant to platelet responses to thrombin, with little effect on thrombus formation in response to collagen.

Both PDI and Nox-1 contribute to oxidative stress observed in individuals with cardiometabolic risk factors. Indeed, we have previously proposed that platelet PDI could be implicated in platelet hyperactivation found in obesity [20]. In addition, PDI expression has been shown to be elevated in atherosclerotic lesions [12], while Nox-1 is reported to be upregulated in mesenteric arteries of mice with metabolic syndrome and contributes to vasodilation of these vessels [21]. Despite evidence in vascular cells, the potential relationship between PDI and Nox-1 biochemistry and cardiometabolic risk factors has not been explored. It is also unknown if these proteins act through similar mechanisms to regulate platelet activation.

Given the individual roles of PDI and Nox-1 in the control of healthy platelets and oxidative stress associated with cardiometabolic risk factors, we sought to examine if these proteins collaborate to control platelet function and if PDI and Nox-1 are associated with cardiometabolic risk factors. Here we show that the co-inhibition of these proteins led to a GPVI-specific additive anti-platelet effect, suggesting that PDI and Nox-1 regulate different pathways downstream of GPVI. Indeed, in a population study, platelet PDI levels were found to be upregulated in obesity, while platelet Nox-1 was increased in obesity, central obesity and in individuals with high blood pressure. These data indicate that dual inhibition of PDI and Nox-1 is a promising strategy for cardioprotection in individuals with cardiometabolic risk factors.

## 2. Materials and Methods

### 2.1. Washed Platelets Preparation

Platelet-rich plasma (PRP) and washed platelets (WP) were prepared from freshly donated blood from consenting healthy donors exactly as previously described [22]. For more details, please refer to Appendix A.

### 2.2. Collection of Mouse Blood and Platelet Preparation

Nox-1^−/−^ mice originally described by Gavazzi et al. [23] were purchased from Jackson Laboratory (Sacramento, CA, USA) and C57BL/6 were used as controls, as recommended by the animal provider. Animals were kept under a 12 h light cycle, controlled temperature (22–24 °C) and food and water ad libitum. Mice (11–14 weeks, females) were culled by rising CO_2_ concentration and blood collected through cardiac puncture into a syringe containing 3.2% sodium citrate at a 1:9 *v*/*v* citrate-blood ratio. Whole blood was centrifuged at 203× *g* for 8 min and PRP collected. 1.25 μg/mL PGI_2_ was added and PRP centrifuged at 1028× *g* for 5 min and pellet resuspended in modified Tyrode’s-HEPES buffer (THB) to obtain WP.

### 2.3. Immunofluorescence Microscopy

Human PRP was activated in the presence of 2 µg/mL integrilin and fixed in 5% paraformaldehyde. Platelets were washed in 1:9 *v*/*v* ACD-phosphate buffer solution (PBS) and left to adhere onto poly-L-lysine coverslips for 90 min. Coverslips were blocked with 1% *w*/*v* bovine serum albumin (BSA) diluted in PBS and incubated with primary or IgG control antibodies overnight at 4 °C. Antibodies were washed away with PBS and secondary antibodies tagged with appropriate fluorophores added for 1 h at room temperature. Finally, coverslips were mounted in gold anti-fade onto slides and analysed with a 100 × magnification oil-immersion lens on a Nikon A1-R confocal microscope (Nikon Optical, Milton Keynes, UK). Anti-PDI (NB600-1164, clone RL77) and anti-Nox-1 (NBP1-31546) were purchased from Novus Biologicals (Bio-techne R&D Systems Europe Ltd., Abingdon, UK)

### 2.4. Turbidimetry and Plate-Based Platelet Aggregation

Turbidimetry [22] and plate-based platelet aggregation [24] were performed using standard protocols as described previously. For more details, please refer to Appendix A.

### 2.5. Fibrinogen Binding and P-Selectin Exposure

Human WP (4 × 10^8^ platelets/mL) were incubated with inhibitors for 10 min. Platelets were then activated with collagen-related peptide (CRP) [25] for 10 min and incubated with 1:50 *v*/*v* FITC-conjugated fibrinogen or 1:50 *v*/*v* PE/Cy5-conjugated anti-human CD62P for 30 min. Events were acquired using a BD Accuri C6 plus flow cytometer.

### 2.6. Calcium Measurement

Human PRP was incubated with 2 µM Fura-2 AM for 1 h at 30 °C. PRP was centrifuged at 350× *g* for 20 min and platelets (4 × 10^8^ platelets/mL) resuspended in THB. Platelets were transferred to a 96-well black plate with clear bottom and incubated with Bepristat (selective PDI inhibitor, [26]) and/or ML171 (selective Nox-1 inhibitor, [27] and [28]) for 10 min and stimulated with 1 µg/mL CRP. Fluorescence was read every 5 s for 5 min using a Flexstation 3 fluorimeter (excitation 340 and 380 and emission 510 nm). Calcium signals were derived from the emission ratios.

### 2.7. Platelet Spreading

Human WP (2 × 10^7^ platelets/mL) were incubated with inhibitors or vehicle for 10 min and left to adhere to collagen-, fibrinogen- or CRP-coated surfaces for 45 min at 37 °C. Non-adherent platelets were washed off three times with PBS. Paraformaldehyde (0.2% *v*/*v*) was added for 10 min to fix the platelets. Triton-X 0.01% *v*/*v* was added for 5 min to permeabilize the cells. Platelets were stained with Alexa Fluor 488-conjugated phalloidin (1:1000 *v*/*v*) for 1 h in the dark at room temperature and analysed using a 20× objective lens on a Nikon A1-R Confocal microscope.

### 2.8. Immunoblotting

Immunoblots were performed using standard protocols as previously described [22]. For details, please refer to Appendix A.

### 2.9. Tail Bleeding Assay

Nox-1^−/−^ or C57BL/6 wildtype (WT) mice were anaesthetised through an intraperitoneal injection of ketamine (100 mg/kg) and xylazine (10 mg/kg). After animals were fully anaesthetised, Bepristat (0.5 µL of a 100 µM solution diluted in 100 µL PBS per 25 g of animal; 50 µM in vivo concentration) was injected intravenously. Tail bleeding was performed as described previously [22] and is detailed in Appendix A.

### 2.10. Population Study

This study comprised of 136 volunteers aged 30 to 65 not using chronic medications that were recruited at the University of Reading to assess physical, metabolic and platelet characteristics. Volunteers answered a questionnaire about their age, gender, amongst other questions not included in this study. Height, weight, body mass index (BMI), blood pressure (BP, measured seated with an electronic automatic sphygmomanometer), and waist and hip circumferences were measured in each volunteer. Blood was taken after overnight fasting and serum glucose levels measured using standard biochemistry protocols. Platelets were washed and immunoblotting performed as above. A protein loading control—GAPDH was used to normalize levels of PDI and Nox-1 to protein loading in each well.

Volunteers were stratified according to their BMI as healthy weight (18.5–24.9 kg/m^2^), overweight (25–29 kg/m^2^), class 1 obesity (30–34.9 kg/m^2^) and class 2 obesity (35–39.9 kg/m^2^). BP was stratified according to the International Society of hypertension [29]: normal (systolic < 130 and diastolic < 85 mmHg), high-normal (systolic 130–139 and/or diastolic 85–89 mmHg), grade 1 hypertension (systolic 140–159 and/or diastolic 90–99 mmHg) and grade 2 hypertension (systolic ≥ 160 and/or diastolic ≥ 100 mmHg). Glycaemia was stratified according to the American Diabetes Association [30]: normoglycaemia (<5.6 mmol/L), impaired fasting glycaemia (IFG) (5.6–6.9 mmol/L) and hyperglycaemia (>6.9 mmol/L). Waist circumference was stratified according to the European Society of Cardiology [31]: normal (Caucasian men < 94 cm; men of other ethnicities < 90 cm; women < 80 cm) and central obesity (Caucasian men ≥ 94 cm; men of other ethnicities ≥ 90 cm; women ≥ 80 cm).

### 2.11. Statistical Analysis

Statistical analyses were performed on GraphPad Prism 8.0 software (GraphPad Software, San Diego, CA, USA). Bar graphs and tables express mean ± SEM. Sample size varied from 4–6 independent experiments for in vitro experiments and between 6 and 8 for tail bleeding experiments. Outliers were determined and excluded by ROUT test. For in vitro experiments using inhibitors, statistical analysis was performed through paired one-way ANOVA and Tukey as post-test, whereas for in vivo experiments using Nox-1^−/−^ mice, these were analysed through two-way ANOVA and Sidak’s multiple comparisons test.

For the population study, linear regression was used to assess the correlation between platelet PDI and Nox-1 levels. To assess the possible association of platelet Nox-1 and PDI with risk factors for cardiac events, volunteers were stratified according to their BMI, BP, waist circumference and glycaemia. Analysis was performed through unpaired one-way ANOVA and Tukey as post-test.

### 2.12. Study Approval

University of Reading Research Ethics Committee approved the population study design and all protocols to obtain and use human blood samples. All participants provided informed consent to participate in this study (AM021702). The University of Reading Local Animal Welfare and Ethics Research Board approved all protocols within a license issued by the British Home Office (I84CBCE3F).

## 3. Results

### 3.1. PDI and Nox-1 Cellular Localization in Resting and Activated Platelets

Since both PDI and Nox-1 are able to regulate ROS generation [12], we would anticipate them to be present in a similar location, and close to their site of activity. Indeed, PDI has been reported to be able to associate with p47phox, which is a cytosolic subunit that regulates the activation of Nox-1 [12]. Therefore, we initially assessed if PDI, p47phox and Nox-1 would co-localise in platelets using immunofluorescence microscopy (Figure 1). Co-localisation was assessed calculating Pearson’s coefficient. In resting, platelets, PDI (blue) and p47phox (red) were localized primarily in the intracellular space, while Nox-1 staining was found to be in ring-like structures, suggesting that it localises on the plasma membrane. CRP activation increased the apparent level of co-localisation of PDI and p47phox with Nox-1 (Figure 1C). However, there was a decrease in the co-localisation of PDI and p47phox upon CRP (a GPVI-specific ligand) activation, suggesting that these proteins may translocate to different compartments in activated platelets. Although we were unable to ascertain if PDI and Nox-1 interact, these data suggest that PDI and p47phox migrate to different compartments in CRP-activated platelets.

### 3.2. Co-Inhibition of PDI and Nox-1 Results in Additive Inhibitory Effect on Platelet Aggregation Induced by Collagen and CRP

Since (1) PDI and p47phox dissociate upon CRP-induced platelet activation and (2) p47phox associates and activates Nox-1, we hypothesized that PDI and Nox-1 regulate distinct processes in platelets. We therefore assessed the effect of inhibition of either, or both proteins together to ask whether outcomes were additive. Platelet aggregation was measured using a high-throughput end-point assay (Figure 2). Neither 3 μM ML171 (Nox-1 inhibitor, also known as 2-APT) nor 15 μM bepristat (selective PDI inhibitor) significantly reduced platelet aggregation by themselves in our experimental conditions, in spite of a visible trend (Figure 2A,B). However, co-incubation with bepristat and ML171 led to an additive inhibitory effect in collagen and CRP-stimulated platelets (Figure 2A,B), but not in TRAP-6 or PMA-stimulated platelets (Figure 2C,D), consistent with the selective role of Nox-1 in GPVI-mediated platelet aggregation [19]. Of note, TRAP-6 and PMA were used to assess if PDI and Nox-1 would inhibit platelet aggregation induced by pathways that are not dependent on GPVI or collagen. Other thiol isomerase inhibitors co-incubated with ML171 yielded a similar additive inhibitory effect in collagen-stimulated platelets (Appendix A). A lower concentration of ML171 was used due to different experimental conditions required for different assays (Appendix A).

### 3.3. Co-Inhibition of PDI and Nox-1 Leads to an Additive Inhibitory Effect on Platelet Activation and Calcium Mobilization Induced by CRP

Individually, both bepristat and ML171 showed inhibitory effects in fibrinogen binding and P-selectin exposure (Figure 3A,B). When co-incubated these inhibitors potentiated the response and caused a ~60% decrease in fibrinogen binding and P-selectin exposure (Figure 3A,B). Calcium mobilization was reduced by 33% when bepristat and ML171 were co-incubated; an effect not seen when each inhibitor was added alone (Figure 3C,D). Of note, a lower concentration of bepristat was used for calcium mobilization due to complete inhibition at higher concentrations (data not shown). No additive inhibitory effect was observed on platelet spreading on surfaces coated with collagen, CRP or fibrinogen (Appendix A). These data indicate that bepristat and ML171 have an additive effect on GPVI-evoked activation of platelets.

### 3.4. Co-Inhibition of PDI and Nox-1 Disrupts Collagen-Stimulated Signalling

Immunoblot analysis was performed to determine the effect of PDI and/or Nox-1 inhibition on several components of the collagen-stimulated signalling pathway, to establish the mechanistic basis of single and dual inhibition. Initially we assessed receptor-proximal molecules such as Src and Syk, however, no changes were detected (Appendix A). Tyrosine phosphorylation, which is integral to platelet signalling pathways that control platelet activation, particularly on stimulation with collagen, was assessed through immunoblot analysis, and was also unaffected (Appendix A). Likewise, bepristat and ML171 exerted no effect on VASP phosphorylation, which mediates powerful inhibitory responses to nitric oxide and PGI_2_ (Appendix A). Therefore, we focused on proteins regulated by ROS that signal further downstream following activation of collagen receptor signalling, analysing platelets stimulated with collagen and lysed at 90 s (Figure 4). ML171 alone was able to decrease the phosphorylation of Akt_S473_, while bepristat alone inhibited the phosphorylation of ERK_T202/Y204_. When co-incubated, bepristat and ML171 caused a ~70% decrease in Ser phosphorylation of putative PKC substrates and inhibited Akt_S473_, p38_T180/Y182_ and ERK_T202/Y204_ to a similar extent. Interestingly, ML171 and bepristat were also able to decrease phosphorylation of p47phox_S370_ (Appendix A). Together, PDI and Nox-1 co-inhibition disrupted the activation of classical platelet-activating pathways, namely PKC, Akt, ERK and p38MAPK, suggesting that PDI and Nox-1 act at parallel pathways to regulate the GPVI signalling pathway. A scheme summarizing the molecular processes regulated by PDI and Nox-1 in collagen-mediated signalling is presented in Appendix A.

### 3.5. Anti-Platelet Effects in Nox-1^−/−^ Mice Are Increased by PDI Inhibition Without Increasing Bleeding Time

Nox-1^−/−^ mice were used to further explore the contribution of Nox-1 and PDI to the regulation of platelet function. Full blood counts are provided in Appendix A and show that Nox-1^−/−^ mice had a ~20% increase in circulating leukocytes and lymphocytes when compared to those of WT. Nox-1^−/−^ platelets treated with bepristat displayed diminished platelet aggregation and putative PKC substrate phosphorylation; an effect not seen when platelets from WT mice were treated with the same concentration of bepristat (Figure 5A–C). Of note, a lower concentration of bepristat was used in experiments with murine platelets, since higher concentrations led to full inhibition (data not shown). This could be due to a higher sensitivity of murine platelets to PDI inhibition when compared to human cells. There was no change on total protein tyrosine phosphorylation (Figure 5D), corroborating data using human platelets (Appendix A). Despite defects in platelet function and signalling, neither Nox-1 deletion nor treatment with bepristat (or both) affected bleeding times (Figure 5E), suggesting Nox-1 and PDI co-inhibition does not affect primary haemostasis substantially. It should be noted that the concentration of bepristat used for tail bleeding experiments was based on the original paper that described the use of this inhibitor [26].

### 3.6. Platelet PDI and Nox-1 Protein Levels Are Upregulated in Conditions of Increased Cardiovascular Disease Risk

After establishing that the co-inhibition of PDI and Nox-1 is a potential strategy to inhibit platelets with minimal bleeding, we evaluated if the expression levels of these proteins would be correlated with conditions of increased cardiovascular risk, i.e., obesity, central obesity, high blood pressure and hyperglycaemia, where elevated platelet function has been noted [32,33]. This assessment is important, since individuals within these subgroups present higher risk of developing cardiovascular events [34]. Therefore, the levels of both proteins were measured with several metabolic characteristics in 136 individuals not in use of long-term medications (Figure 6) (for descriptive statistics, see Appendix A). There was no correlation between protein levels of PDI and Nox-1 in platelets (Figure 6A), indicating that different pathways regulate the expression levels of these proteins. Platelet PDI was upregulated in obesity (25% higher in class 2 obesity vs. healthy weight) (Figure 6B), while Nox-1 was increased in individuals that presented with obesity (75% higher in class 2 obesity vs. healthy weight), high blood pressure (25% higher in high blood pressure stage 2 vs. normal) or central obesity (25% higher in central obesity vs. normal) (Figure 6E–G). Platelet levels of these proteins were unchanged in hyperglycaemia (Appendix A).

## 4. Discussion

Data herein presented suggest that platelet PDI and Nox-1 are correlated with cardiometabolic risk factors and dual inhibition may provide a new pharmacological avenue for the development of anti-platelet medications. Using selective inhibitors in both human and murine Nox-1^−/−^ platelets, it was established that the co-inhibition of PDI and Nox-1 led to an additive inhibitory effect on GPVI-mediated platelet aggregation, activation, calcium flux and signalling. Moreover, no bleeding defect was observed in Nox-1^−/−^ mice treated with the PDI inhibitor bepristat, suggesting the anti-platelet effects were not correlated with defects in haemostasis.

In VSMC, PDI has been reported to regulate Nox-1 activity [35]. This involves the formation of a disulphide bond between Cys400 of PDI and Cys196 of p47phox, a cytosolic subunit of the Nox-1 complex [12]. In leukocytes, reduced PDI was shown to associate with p47phox and to dissociate upon activation with PMA when compared to resting leukocytes [36]. Our data suggest a migration of both PDI and p47phox towards the platelet membrane, where the active Nox-1 complex is located, upon activation with CRP, which is consistent with PDI and Nox-1 co-localization in HEK293 cells [37]. Indeed, PDI is highly expressed and trapped in the dense tubular system (DTS) [8], while little is known of where p47phox is localized in platelets. In parallel, PDI and p47phox dissociated upon platelet activation with CRP, suggesting that these proteins migrate to different compartments in activated platelets or that PDI remains trapped in the DTS while p47phox translocates to the membrane. Further experiments are needed to ascertain how PDI and p47phox interact in platelets.

Platelets pre-treated with both PDI and Nox-1 inhibitors tended to aggregate, activate and signal less than platelets treated with the two inhibitors used separately. This was confirmed using different PDI inhibitors and in murine Nox-1 deficient platelets incubated with bepristat. Although it is possible that the effects observed could be due to partial PDI inhibition by bepristat, this is unlikely since it was shown that 10 μM bepristat abrogated PDI activity measured through the insulin turbidimetry assay [26]. Moreover, it is also possible that other thiol isomerases, such as ERp5, ERp57 and ERp72, could cooperate with Nox-1 to regulate platelets, since (1) these thiol isomerases were shown to modulate platelet function and thrombus formation [38] and (2) ERp72 was reported to co-localize with and modulate the function of Nox-1 in CaCO-2 and Cos-1 cell-lines [39]. The potential synergism between other thiol isomerases and Nox-1 will be further assessed in the future.

Nonetheless, inhibition of PDI in Nox-1^−/−^ mice did not affect bleeding time. Previous reports have shown that neither PDI deletion [6] nor Nox-1 deletion [27] increase bleeding time In line with our data, previous reports have shown that platelets from Nox-1 KO mice did not respond to collagen, whilst responses to thrombin were preserved [19]. It has also been reported that ML171 (also referred to as 2APT) does not affect platelet adhesion, nor does it inhibit aggregation induced by agonists other than collagen [18,40]. In contrast, one study reported that male Nox-1 KO mice presented defects in thrombin-stimulated platelets [17]. The reason for this inconsistency is unclear, although a difference in the sex of the mice utilised in this and previously quoted studies may be the cause of this discrepancy. Since the Nox-1 gene is located on chromosome X, sexual dimorphism is a possibility. Nonetheless, our data using female Nox-1^−/−^ mice are in agreement with the majority of the literature, and propose for the first time that PDI and Nox-1 both contribute to the regulation of platelet responses in a GPVI-specific manner.

To explore the mechanisms underlying the anti-platelet effects of PDI and Nox-1 dual inhibition, well-characterised platelet protein kinase-dependent pathways were investigated. Our data show that 90 s after collagen addition, ML171 alone was able to decrease Akt phosphorylation, while bepristat alone reduced the phosphorylation of ERK1/2. The PI3K-Akt signalling pathway is a known regulator of GPVI-induced platelet activation and granule secretion [41]. Indeed, deletion of Akt1 diminishes collagen-induced platelet activation [42]. Similar to the PI3K-Akt pathway, MAPKs, such as ERK1/2, are activated downstream of various agonists, while selective inhibition of ERK1/2 modulates collagen-induced platelet responses (recently reviewed in [43]). Interestingly, ERK1/2 and p38 MAPK were shown to have complementary effects to regulate platelet function downstream of collagen [44]. These results suggest that Nox-1 is involved in PI3K signalling pathway, while PDI can regulate MAPKs downstream of collagen.

Interestingly, both Akt [45] and ERK1/2 [43] were shown to be involved in PKC signalling in platelets. Indeed, PKC is a signalling hub downstream of several platelet receptors that can also induce p47phox phosphorylation, and thus activate Nox-1 [46]. Therefore, we investigated the effect of dual inhibition of PDI and Nox-1 on the phosphorylation of PKC substrates, p38 MAPK and p47phox, in addition to Akt and ERK1/2. Dual inhibition of PDI and Nox-1 decreased phosphorylation levels of PKC substrates, p38 MAPK, ERK1/2, Akt and p47phox. Altogether, we propose that regulation of the MAPKs and PI3K pathways by PDI and Nox-1, respectively, may sustain signalling downstream of collagen and dual inhibition of PDI and Nox-1 may affect other molecules, such as PKC and p47phox, that lead to an additive inhibitory effect of platelet function.

Of note, the Nox complex was postulated to be constitutively associated to the cytoplasmic tail of GPVI and its assembly may comprise an initial step of GPVI-induced platelet activation [4]. In juxtaposition, it is unlikely that the additive effect of PDI and Nox-1 co-inhibition is due to PDI regulation of integrins [6], given that (1) the additive inhibitory effects on platelet function were specific to GPVI ligands and (2) platelet adhesion, which is mainly regulated by integrins, was not affected. Nevertheless, PDI inhibition in platelets of Nox-1^−/−^ mice did not completely block aggregation or signalling induced by collagen, suggesting that additional underlying pathways may sustain collagen-induced responses. These pathways, such as second messengers and activation of different kinases, should be further explored in the future.

PDI and Nox-1 were shown to be upregulated in resistance arteries of hypertensive mice [47], as well as in human atheromatous plaques [12]. While Nox-1 has been implicated in the aetiology of obesity and metabolic syndrome [48], PDI could be relevant in platelet hyperactivation of obese subjects [20]. Indeed, circulating levels of PDIA4 were positively associated with metabolic syndrome in adults [49]. Here we provide the first human study of a positive association between platelet PDI and Nox-1 in obese subjects. Moreover, platelet Nox-1 levels were increased in individuals presenting high blood pressure and central obesity. It is therefore possible that PDI and Nox-1 may be implicated in the pathophysiology of platelet hyperactivation observed in obesity and hypertension (reviewed in [32,33]). Underlying mechanisms of the upregulation of Nox-1 and PDI may include oxidative stress and endoplasmic reticulum stress, which are well-established components in the pathophysiology of metabolic and cardiovascular dysfunction [2,50]. Indeed, Nox-1 was described to be upregulated and to mediate ROS generation in vessels of hypertensive rats [51], whilst PDI is an essential redox regulator of the unfolded protein response associated with endoplasmic reticulum stress [52]. Altogether, the upregulation of platelet PDI and Nox-1 in conditions of increased cardiometabolic risk suggests that these proteins could be involved in the platelet dysfunction observed in obese and hypertensive individuals. The underlying mechanisms of these observations should be investigated in the future.

In conclusion, the co-inhibition of platelet PDI and Nox-1 resulted in decreased platelet aggregation, fibrinogen binding, P-selectin exposure, calcium mobilization and signalling. Expression levels of PDI and Nox-1 were independent of one another in platelets, suggesting PDI and Nox-1 may act at different points in the GPVI signalling pathway. Finally, platelet PDI and Nox-1 were upregulated in conditions of increased cardiovascular risk. Altogether, we propose that PDI and Nox-1 are promising targets for the development of new anti-platelet strategies as secondary prevention in chronic metabolic diseases. Future studies should address the pathophysiological relevance of these proteins in platelets to the development of cardiac events in patients at high risk.

## Figures and Tables

**Figure 1 antioxidants-10-00497-f001:**
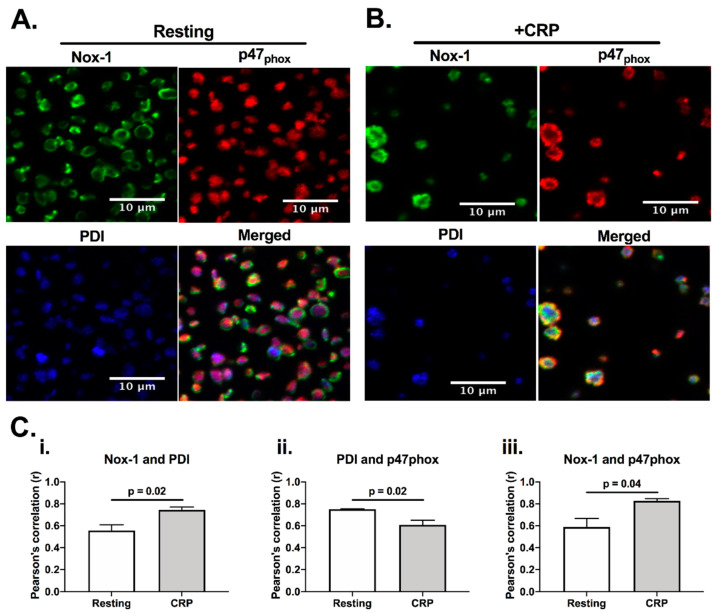
PDI and Nox-1 cellular localization in resting and CRP-activated platelets. Immunofluorescence of platelet-rich plasma (PRP) not stimulated (resting) (**A**) or stimulated with 1 μg/mL CRP (**B**) for 3 min in the presence of 4 μg/mL integrilin. (**C**) Pearson’s correlation of at least 3 different fields for 3 independent experiments represents the degree of co-localization of Nox-1 and PDI (**i**), PDI and p47phox (**ii**) and Nox-1 and p47phox (**iii**). Bar graphs show mean ± SEM. Data analysed by paired Student *t*-test. Exact *p*-value are presented in each bar graph. Pink colour is the colocalisation of p47phox (red) and PDI (blue), yellow is the colocalisation of p47phox and Nox-1 (green) and white is the colocalisation of all three proteins.

**Figure 2 antioxidants-10-00497-f002:**
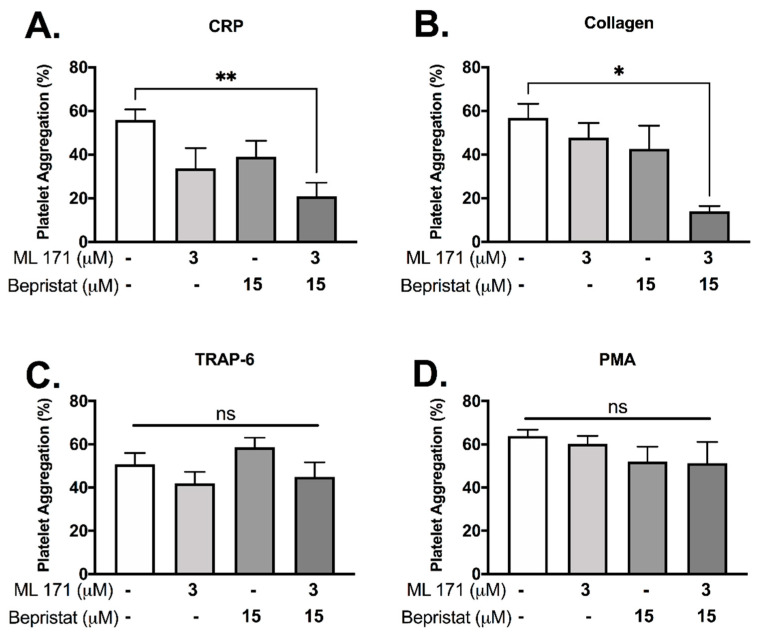
Bepristat and ML171 exert an additive inhibitory effect on plate-based platelet aggregation induced by GPVI agonists. Human WP at 4 × 10^8^ platelets/mL were pre-treated with 3 µM ML171 and/or 15 µM Bepristat for 10 min prior to addition of agonists: 1 µg/mL CRP (**A**), 2 µg/mL Collagen (**B**), 10 µM TRAP-6 (**C**) or 500 nM PMA (**D**) (*n* = 4–5). Platelet aggregation was measured using a plate-based assay. Data on graphs show mean ± SEM. Data analysed by paired one-way ANOVA and Tukey’s multiple comparisons test. * *p* < 0.05 and ** *p* < 0.01. ns: non-significant.

**Figure 3 antioxidants-10-00497-f003:**
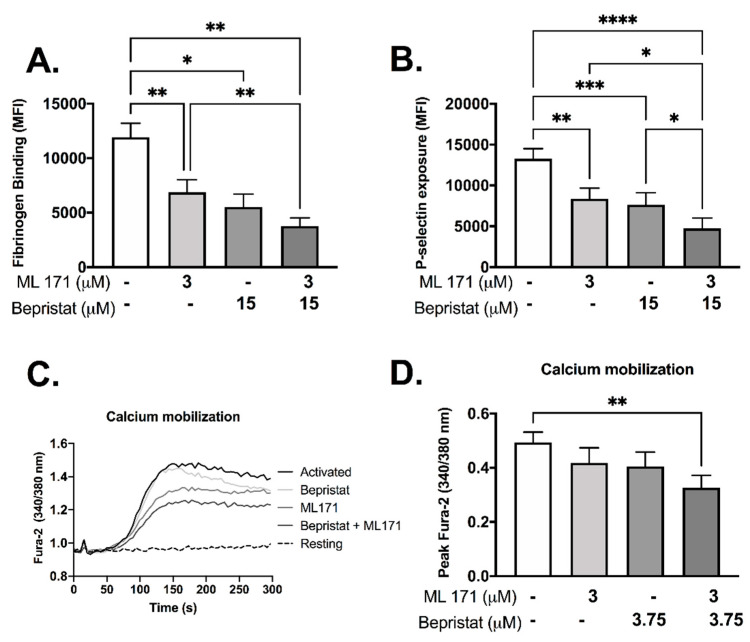
Bepristat and ML171 display additive inhibitory effects on CRP-induced fibrinogen binding, P-selectin exposure and calcium mobilization. (**A**,**B**): Human WP (4 × 10^7^ platelets/mL) were incubated with 3 µM ML171 and/or 15 µM bepristat for 10 min, then 1 µg/mL CRP was added. FITC-fibrinogen and PE/Cy5-anti-P-selectin were incubated for 30 min and events acquired using a flow cytometer. Data expressed as median fluorescence intensity (MFI). (**C**): Representative curve of WP pre-incubated with calcium dye Fura-2 AM and pre-treated with 3 µM ML171 and/or 3.75 µM bepristat for 10 min prior to activation with 1 µg/mL CRP. Fluorescence acquired over 5 min using a plate reader. (**D**) Summary statistics for data in (**C**) activated with CRP. Peak Fura-2 was determined as the highest fluorescence value subtracted from baseline before agonist addition. *n* = 6 for (**A**,**B**), while *n* = 4 for (**C**,**D**). Bar graphs show mean ± SEM. Data analysed by paired one-way ANOVA and Tukey’s post-test. * *p* < 0.05, ** *p* < 0.01, *** *p* < 0.001 and **** *p* < 0.0001.

**Figure 4 antioxidants-10-00497-f004:**
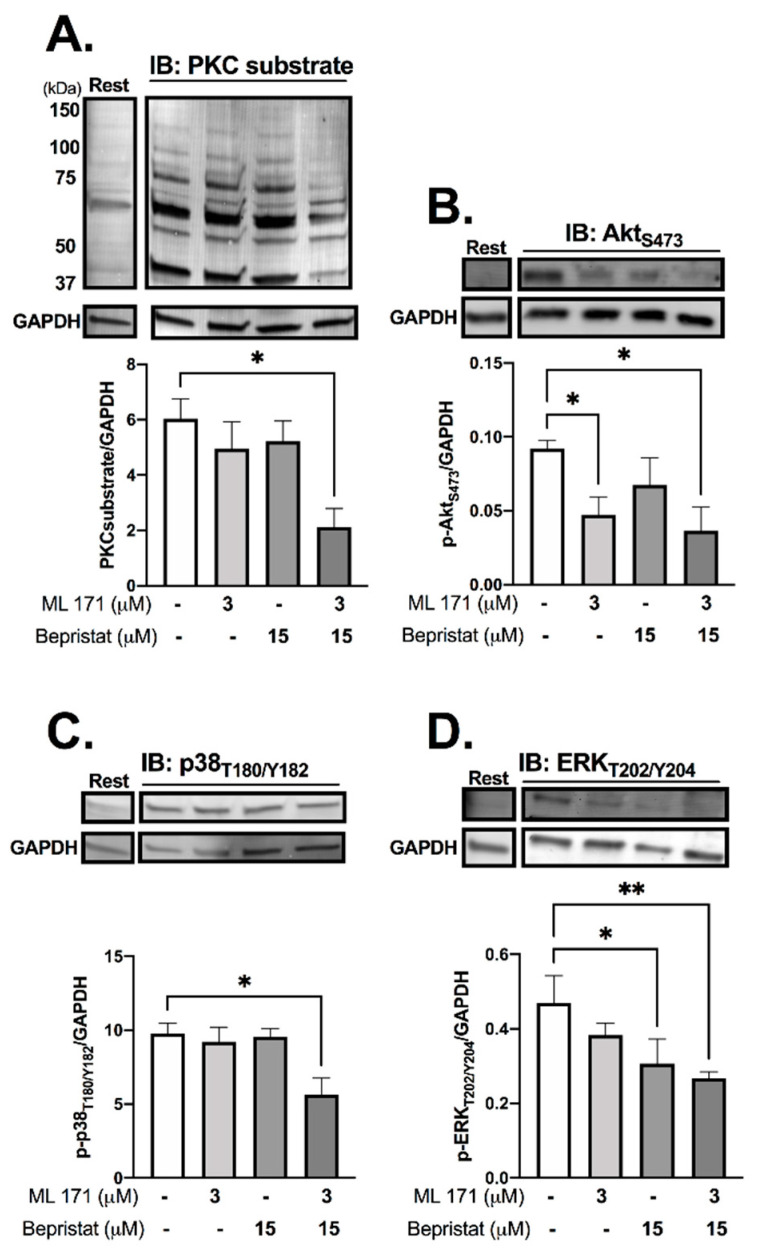
PDI and Nox-1 co-inhibition decreases phosphorylation of PKC substrates, Akt, p38MAPK and ERK. WP at 4 × 10^8^ platelets/mL were incubated with 3 µM ML171 and/or 15 µM bepristat for 10 min prior to adding 3 µg/mL Collagen. Platelets were lysed after 90 s and immunoblots performed. Samples were tested for: putative PKC substrate phosphorylation (**A**), AktS473 (**B**), p38 MAPKT180/Y182 (**C**) and ERKT202/Y204 (**D**). GAPDH was used as a control for equal loading. Representative blot is presented above of bar graphs with summary statistics. Each lane represents the condition in graph below. A basal unstimulated condition is presented (Rest). Data are representative of 3–4 independent experiments. Bar graphs show mean ± SEM and were analysed by paired one-way ANOVA and Tukey’s post-test. * *p* < 0.05, ** *p* < 0.01.

**Figure 5 antioxidants-10-00497-f005:**
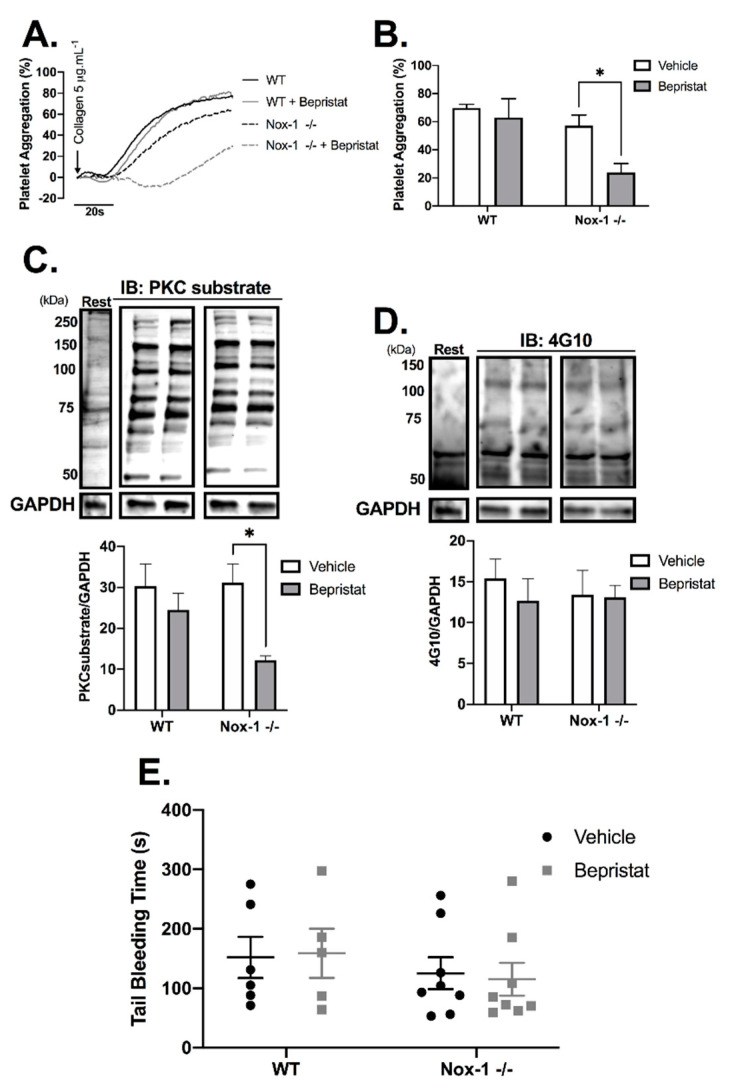
Nox-1^−/−^ mice treated with bepristat showed additive anti-platelet effects with no repercussion on bleeding time. (**A**) Platelet aggregation curves of mouse wildtype (WT) and Nox-1^−/−^ platelets pre-treated with 7.5 µM bepristat or vehicle for 10 min and activated with 5 µg/mL Collagen. (**B**) Summary statistics of aggregation curves. Immunoblots were performed in platelets pre-treated with 7.5 µM bepristat or vehicle for 10 min and activated with 5 µg/mL Collagen for 90 s. (**C**) PKC substrate. (**D**) 4G10 total Tyr phosphorylation. Representative blots are presented on top of bar graphs with summary statistics. Each lane represents the condition in graph below. A basal unstimulated condition is presented (Rest). (**E**) Tail bleeding time in mice injected with 50 µM bepristat or vehicle control. *n* = 5–6 for (**A**–**D**), while *n* = 5–8 for (**E**). Data on graphs expressed as mean ± SEM analysed by unpaired two-way ANOVA and Sidak’s post-test. * *p* < 0.05.

**Figure 6 antioxidants-10-00497-f006:**
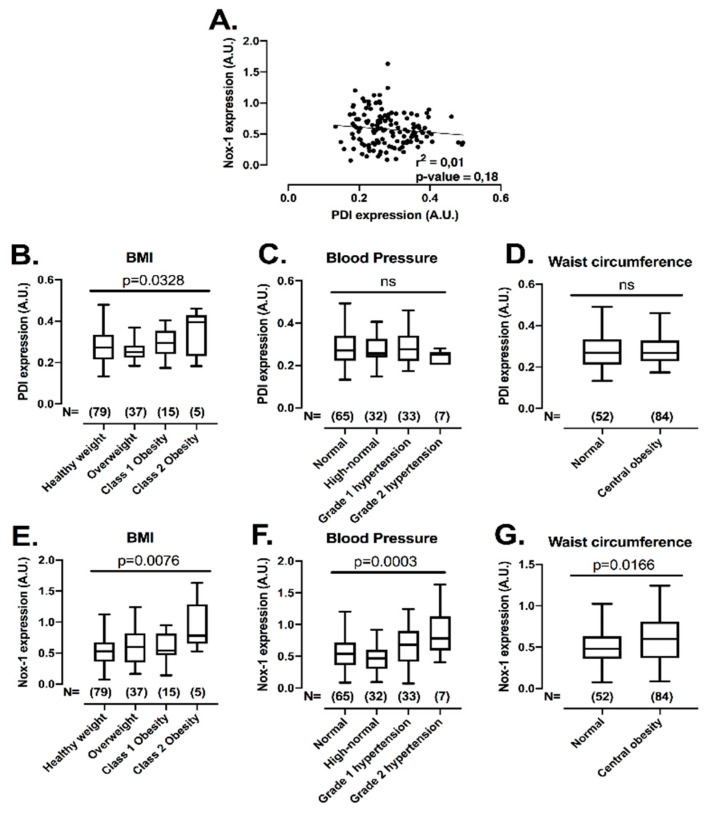
Platelet PDI and Nox-1 levels are independent of one another and upregulated in conditions of increased cardiovascular risk. Washed platelets (WP) from 136 volunteers were lysed and immunoblots performed for PDI, Nox-1 and loading control GAPDH. Anthropometric and metabolic characteristics were also collected. Value cut-offs of stratifications can be found on Methods and were all performed according to international guidelines. (**A**) Linear regression of platelet PDI and Nox-1 protein levels. (**B**–**G**) PDI and Nox-1 expression were stratified by: BMI (**B**,**E**) in healthy weight (18.5–24.9 kg/m^2^), overweight (25–29 kg/m^2^), class 1 obesity (30–34.9 kg/m^2^) and class 2 obesity (35–39.9 kg/m^2^); blood pressure (**C**,**F**) in normal (<120/80 mmHg), elevated (120–129/80 mmHg), high blood pressure (HBP) stage 1 (130–139/80–89 mmHg), HBP stage 2 (140/90 or above mmHg); waist circumference (**D**,**G**) in normal (Caucasian men < 94 cm; men of other ethnicities < 90 cm; women < 80 cm) and central obesity (Caucasian men ≥ 94 cm; men of other ethnicities ≥ 90 cm; women ≥ 80 cm). Data in graph show box and whiskers depicting median, range and 25th and 75th percentiles analysed by one-way ANOVA and Tukey’s post-test. Overall *p*-value of one-way ANOVA is shown. ns: non-significant.

## Data Availability

Data will be made available upon request to the corresponding author.

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
