# Peer review of "Protein Disulphide Isomerase and NADPH Oxidase 1 Cooperate to Control Platelet Function and Are Associated with Cardiometabolic Disease Risk Factors"

_antioxidants, 2021, doi:10.3390/antiox10030497_

Round 1

Reviewer 1 Report

COMMENTS TO THE AUTHORS

This is a well planned, well executed research dealing with an important aspect of platelet function. The manuscript is well and clearly written. The Introduction is comprehensive and focused. The methodology is detailed and allows a proper understanding of the results. The Discussion is balanced and emphasizes the take-home message that PDI and Nox1 are promising targets for drugs able to dampen excessive platelet activity.

The authors were courageous in engaging in a study that comprises in vitro experiments with human and mouse platelets, in vivo experiments in mice, and work with human cells, derived from people with increased cardiovascular disease risk.

Points to be addressed by the authors:

  1. Protein disulfide isomerases (PDIs) are a large family of proteins. PDIA1, ERp5 and ERp57 were implied in platelet function (see role of ERp57 in platelet aggregation in Y. Wu et al., Blood 119, 1737-1746, 2012). Can the authors provide information on the particular PDI dealt with in this study.
  2. Please indicate the particular antibodies used for the immunofluorescence experiments described in Fig. 1 (for Nox1, PDI, p47-phox). Specificity of antibodies to various forms of PDI and to Nox1 is not a trivial issue and should be addressed (antibodies to p47-phox represent no problem).
  3. Showing the pictures and graphs in Fig. 1 is interesting and justified but I would urge the authors to exhibit moderation in the interpretation of the data. Thus, there seems to be little difference in the localization of Nox1 in resting and activated platelets (most of it is in the membrane, though activation seems to enhance membrane localization). Subcellular localization of Nox1 in other cells is still an “open” issue. PDI appears to be cytosolic in both resting and activated platelets and I find the conclusion that PDI migrates to the membrane together with Nox1, but not with p47-phox, upon activation, somewhat strenuous. To me it seems that PDI “stays put” in the cytosol, whereas p47-phox goes to the membrane to link up with Nox1. The fact that the frames of resting platelets are covering a larger number of platelets than those of the activated cells is also elevating the level of uncertainty.
  4. An issue that was not addressed was that, although p47-phox does in certain situations act as a Nox1 organizer, the “physiological “ Nox1 organizer is NOXO1. I was not able to find information in the literature about NOXO1 in platelets. I wonder whether the authors know more about this issue.
  5. This reviewer was not familiar with Pearson’s correlation but on trying to learn about it, it appears to rest on rather complex mathematics (Pearson’s correlation is the covariance of two variables, divided by the product of their standard deviations; thus, it is essentially a normalized measurement of the covariance, such that the result always has a value between −1 and 1). On looking at the graphs in Fig. 1C, one wonders at the “real significance” of the differences in panels i, ii, and iii, in spite of the p values (which are very close to 0.05). Care must be exercised in applying sophisticated statistical formulas to measurements based on visual assessment,
  6. The proposal that PDI and p47-phox dissociate upon platelet activation (based on Fig. 1C, panel ii) is a proposal that should be discussed in connection to the paper by de A. Paes (J. Leukoc. Biol. 90, 799-810, 2011) on the redox-dependent association of PDI with p47-phox in leukocytes.
  7. The Nox1 inhibitor ML171 (2-APT) is not a common chemical and a reference should be provided related to its specificity for Nox1 (D. Gianni et al., ACS Chem. Biol. 5, 981-993, 2010). Please also note paper doubting specificity of ML171 for Nox1 (T. Seredenina et al., Free Rad. Biol. Med. 86, 239-249, 2015).
  8. The PDI inhibitor bepristat was described in 2016 as a member of a family of compounds that inhibit PDI by a specific mechanism affecting the substrate binding region of domain b’. This endows it with specificity for PDIA1 (it does not affect ERp5 and ERp57). The reference to this (R.F. Beckendam et al., Nature Commun. 7:12579, 2016) should be cited.
  9. For the non-initiated, can the authors comment on the lack of effect of Nox1 deletion and PDI inhibition by bepristat on bleeding time.
  10. In the Discussion, the issue of the involvement of different types of PDIs in platelet function should be discussed. Also, the role of extracellular PDIs in platelet thrombus formation should be mentioned. See review: S. Schulman et al., Antiox. Redox.Signal 24, 1-15, 2016.
  11. I noticed that the authors recently published a paper on platelet extracellular vesicles expressing Nox1 (R.S. Gaspar et al., Free Rad. Biol. Med. 165, 395-400, 2021.
  12. A technical point is that, due to the numerous abbreviations, a list of abbreviations would be very helpful.

Author Response

We thank the reviewer for the time to evaluate our work. Your comments were very helpful.

Reviewer 2 Report

This is an interesting work trying to demonstrate the cooperation of PDI and NADPH oxidase 1 to control platelet function. However, there are some concerns:

  1. The authors used different concentrations of agonists and inhibitors in different experiments. For example, usually bepristat at 15 µM concentration was used, but in Fig. 4 at 7.5 µM concentration. Another example is collagen – in Fig. 4B: 2 µg/ml, in Fig. 4: 3 µg/ml and in Fig. 5: 5 µg/ml. The observed final effects can vary dependently on the strength of platelet activation and the strength of platelet inhibition. Can the authors demonstrate concentration-dependent additive effects of ML171 and bepristat on platelet aggregation activated by agonists at different concentrations? Isobologram analysis would be helpful to evaluate drug interactions. Did the authors try to analyse the interaction between ML171 and bepristat by this method? It is also possible that TRAP-6 (10 µM) and PMA (500 nM) are strong enough not to reveal the additive effects of ML171 and bepristat. Why such concentrations were selected? Can the authors demonstrate dose-dependent responses of platelets to agonists and comment on this issue?
  2. Please explain why bepristat in the tail bleeding assay was used at 50 µM concentration? Higher number of platelets in murine in comparison with human blood, as well as in comparison with in vitro assays, may influence the observed effects of bepristat. Did the authors test higher concentrations of bepristat in vivo?
  3. The authors used two methods for the analysis of platelet aggregation: classical turbidimetry and plate-based platelet aggregation. The readers should be informed by the descriptions of the figures, which of these methods was used to obtain the presented data. Additionally, a number of platelets should be always included to make the description self-explanatory, without the necessity to check it in the supplementary data.
  4. It would be beneficial to prepare a scheme summarizing the molecular processes described in the study.
  5. Is it possible that bepristat, by a direct inhibition of GPIIb/IIIa, can affect the phosphorylation of PKC substrates, Akt, p38MAPK and ERK in a mechanism of outside-in signalling from GPIIb/IIIa receptor? Could the authors answer or comment on this question?
  6. Why PMA is used – please explain in the test of the manuscript.
  7. Please define ‘PKC substrates’ – it is not clear.
  8.  

Author Response

Thank you for your time in evaluating our manuscript. Please see the attachment.

Reviewer 3 Report

In this paper, the authors examine the contributions of PDI and Nox1to the regulation of platelet function.  They clearly demonstrate functional roles for PDI and Nox1 in collagen/CRP-stimulated platelets.  However, the roles of PDI and Nox1 in a single or parallel pathways is not as clear.  The most convincing data for parallel pathways are derived from studies on platleet aggregation in Nox1 knockout mice.  These studies also monitored the phosphorylation of PKC substrates, but the PKC experiments require further clarification and improvements to the data presentation.  Other data indicate that calcium mobilization in stimulated platelets can be attributed to a single PDI-dependent signalling pathway, yet the authors minimize this finding in favor of the parallel pathway model.  The authors should consider that both possibilities could be involved in regulating different aspects of platelet function:  for example, a single PDI/Nox1 pathway for calcium mobilization but separate PDI and Nox1 pathways that both contribute to platelet aggregation.  The authors should also consider the possibility that higher drug concentrations may have produced more dramatic (complete?) inhibitory effects, such as seen for the complete inhibition of calcium mobilization with just the PDI inhibitor bepristat.

Major Points

  1. It is appreciated that the authors note a lower concentration of bepristat (3.75 uM) was used in Figure 3C because a complete inhibition of calcium mobilization was recorded with their standard concentration of 15 uM. However, this observation seems to contradict the authors' model that PDI and Nox1 act in parallel pathways for calcium mobilization.  If a strong inhibition of PDI completely blocks calcium mobilization, then there does not appear to be a separate, parallel pathway involving Nox1.  An interpretation of the presented data, with the less effective concentration of bepristat, could be that 3.75 uM bepristat only partially inhibits a single PDI/Nox1 pathway, and the additional partial inhibition of Nox1 with ML171 results in a statistically significant effect.
  2. Did the authors complete dose response curves for ML171 or bepristat? Is it possible that higher concentrations of the single drugs could completely inhibit different aspects of platelet function, as noted for bepristat and calcium mobilization?  Why did the authors choose 3 uM ML171 and 15 uM bepristat for most of their experiments?  
  3. Why was 7.5 uM bepristat used in Figure 5 when it was used at 15 uM concentration in the rest of the paper, with the exception of Figure 3D? Was this because there was a complete inhibition of platelet function with the higher bepristat concentration, as reported for Figure 3D?  In addition, why was 0.75 uM ML171 used in Supplementary Figure 1 when it was used at 3 uM concentration in the rest of the paper?  These differences in drug concentrations could impact the data.  They should be noted in the text, with an explanation (such as given for Figure 3D) for the differences.
  4. The Western blots for "PKC substrates" in Figures 4 and 5 are not clearly explained in the Results or Methods. What are these substrates, and what specific antibody was used to detect them?  There are many proteins detected in these blots, which were quantified in bar graphs as ratios of PKC substrate:GAPDH.  How did the authors quantify all of the proteins in the PKC blot (or did they pick representative bands for quantification?) as a comparison to the single band on the GAPDH blot?  Did every band on the PKC blot exhibit the same proportional decrease in intensity after ML171 or Bepristat treatment?  These are important issues because the intensities of the bands in the gel do not appear to match the quantifications in the bar graphs (especially in Figure 5C).
  5. Why were the Western blots for "PKC substrates" cropped at different locations in Figures 4 and 5? The standards in Figure 4 run from 37 kDa at the bottom of the gel to 150 kDa at the top, but Figure 5 ranges from 50 kDa to 250kDa.  As a result, there are proteins in the Figure 4 gel that are cropped out of the Figure 5 gel and vice-versa.  Is the pattern of PKC substrate phosphorylation that different between human (Figure 4) and mouse (Figure 5) platelets?  It would be best to show the entire blot for both Figures 4 and 5.
  6. Do any of the experiments other than Figure 1 report results for resting cells (ie, no CRP/collagen stimulation)? It would be helpful to compare the baseline signals from resting cells to the signals from stimulated cells and to see that the drug treatments did not affect these resting signals.
  7. For the studies with Nox1 knockout mice (Figure 5A-C), there is still some platelet aggregation and PKC substrate phosphorylation after the inhibition of PDI function. Does this mean there is a third pathway, independent of both Nox1 and PDI?  Did a sub-optimal drug concentration allow some continued PDI activity?  Perhaps the remaining levels of platelet aggregation and PKC substrate phosphorylation are also seen in unstimulated cells, which would then actually represent a complete inhibition of the collagen-stimulated effects?     

Minor Points

  1. Lines 24-25: "PDI and Nox-1 together contributed to GPVI signalling through phosphorylation of p38 MAPK, p47phox, PKC and Akt."  The structure of this sentence makes is sound like PDI and Nox-1 are kinases.  Perhaps replacing "through" with "that involved" would clarify this potential misconception.  Were p47phox and PKC phosphorylation directly demonstrated in this paper?
  2. Line 393: Figure 5I should be Figure 5E
  3. Line 444: Figure 1A should be Figure 6A.
  4. Lines 538-539: "uncoupled protein response" should be "unfolded protein response"
  5. Lines 544-546: In the Discussion, it is stated that "the co-inhibition of platelet PDI and Nox-1 resulted in decreased platelet aggregation, fibrinogen binding, P-selectin exposure, calcium mobilization and signalling, all of which were GPVI-specific."  Only Figure 2, which is focused on platlet aggregation, appears to examine signalling pathways other than GPVI.  Thus, the final part of the sentence ("all of which were GPVI-specific") does not reflect the presented data. 

Author Response

We thank the reviewer for the helpful comments. Please see the attachment.

Round 2

Reviewer 3 Report

My concerns have been addressed in the revision.  I appreciate the effort put into these responses.

Before publication, the authors might consider adding a line to the legends of Figures 4 and 5 that specifically note and define the new "Rest" controls.

Author Response

We thank the reviewer again for evaluating our manuscript. A sentence noting and defining the "Rest" condition was added to the legend of Figures 4 and 5.